# A New Approach for Preparing Stable High-Concentration Peptide Nanoparticle Formulations

**DOI:** 10.3390/ph17010015

**Published:** 2023-12-21

**Authors:** Chloe Hu, Nanzhi Zang, Yu Tong Tam, Desmond Dizon, Kaylee Lee, Jodie Pang, Elizabeth Torres, Yusi Cui, Chun-Wan Yen, Dennis H. Leung

**Affiliations:** 1Synthetic Molecule Pharmaceutical Sciences, Genentech, Inc., 1 DNA Way, South San Francisco, CA 94080, USA; hu.chloe@gene.com (C.H.); zang.nanzhi@gene.com (N.Z.); yen.chunwan@gene.com (C.-W.Y.); 2Pharmaceutical Development, Genentech, Inc., 1 DNA Way, South San Francisco, CA 940802, USA; tam.yu-tong@gene.com; 3Device Development, Genentech, Inc., 1 DNA Way, South San Francisco, CA 94080, USA; dizon.desmond@gene.com; 4Drug Metabolism and Pharmacokinetics, Genentech, Inc., 1 DNA Way, South San Francisco, CA 94080, USA; lee.shi-hui@gene.com (K.L.); pang.jie@gene.com (J.P.); cui.yusi@gene.com (Y.C.); 5Development Sciences, Genentech, Inc., 1 DNA Way, South San Francisco, CA 94080, USA; torres.elizabeth@gene.com

**Keywords:** peptide, nanosuspension, nanoparticle, formulation

## Abstract

The subcutaneous administration of therapeutic peptides would provide significant benefits to patients. However, subcutaneous injections are limited in dosing volume, potentially resulting in high peptide concentrations that can incur significant challenges with solubility limitations, high viscosity, and stability liabilities. Herein, we report on the discovery that low-shear resonant acoustic mixing can be used as a general method to prepare stable nanoparticles of a number of peptides of diverse molecular weights and structures in water without the need for extensive amounts of organic solvents or lipid excipients. This approach avoids the stability issues observed with typical high-shear, high-intensity milling methods. The resultant peptide nanosuspensions exhibit low viscosity even at high concentrations of >100 mg/mL while remaining chemically and physically stable. An example nanosuspension of cyclosporine nanoparticles was dosed in rats via a subcutaneous injection and exhibited sustained release behavior. This suggests that peptide nanosuspension formulations can be one approach to overcome the challenges with high-concentration peptide formulations.

## 1. Introduction

Peptides have emerged as a distinct therapeutic modality compared to typical small-molecule drugs and large-molecule proteins [1,2,3]. However, peptide therapeutics exhibit unique challenges that make their successful development difficult. These can consist of pharmacokinetic liabilities, such as a short half-life and limited absorption across physiological barriers, as well as poor chemical and physical stability [4]. In some cases, both structural modifications as well as formulation development can address these risks to some extent to improve peptide delivery.

In general, protein and peptide drug products are formulated at relatively low concentration for intravenous (IV) administration [5]. On the other hand, subcutaneous (SC) administration would provide significant benefits to patients, including the ability to self-administer the medication at home, thereby reducing costs and increasing compliance. However, SC injections are limited to a dosing volume of 1–1.5 mL, potentially resulting in high peptide dose concentrations of >100 mg/mL [5]. This can result in significant challenges, including solubility limitations, high viscosity, and physical instability, leading to aggregation [6,7]. Higher concentrations often exacerbate these effects and lead to increased risk of peptide–peptide interactions, potentially leading to conformational changes, aggregation, precipitation, or gelation [8]. This can cause loss of activity as well as toxicity and immunogenicity risks [8]. These risks can result in significant difficulties in formulation development.

Peptides are generally preferred to be formulated as solutions in aqueous media [9]. Unlike larger proteins, peptides are smaller and often lack a strong secondary structure. As a result, hydrophobic residues can have a disproportionately large effect on solubility since the lack of a strong secondary structure results in their surface exposure. Additionally, peptides often exhibit sharp pH-dependent solubilities that are difficult to control effectively with buffers. As a result, it can be difficult to formulate peptides at high concentrations in aqueous solutions. In many cases, non-aqueous solvents have been explored, such as the use of ethanol and Cremophor EL in formulating cyclosporine A, although their use remains relatively limited due to their potential for causing unfolding and denaturation as well as having limited pharmaceutical acceptability [10].

In contrast to solutions, stable suspension formulations of peptides in aqueous media may provide an attractive alternative [11]. There have been some recent advances in the use of suspensions for long-acting injectable depot formulations, although these have been largely limited to lipid-based approaches or aqueous suspensions of small molecules [12,13]. Suspension formulation approaches may be able to overcome challenges with solubility limitations when they carried out at higher concentrations. In addition, the chemical stability of the peptide may be improved in its solid state. However, due to their relatively large molecular weight and conformational flexibility, peptides often exist in disordered amorphous forms in their solid state, and control over physical stability and particle size can be difficult. As a result, a general approach for formulating stable peptide suspensions has been lacking.

Thus, there remains a need for a general approach to formulate peptides at high concentrations. Herein, we report on a new approach using resonant acoustic milling [14] for preparing peptide nanoparticle formulations at high concentrations, avoiding the use of high amounts of harsh excipients while enabling good absorption in vivo after a subcutaneous injection. Moreover, while typical milling techniques subject material to high shear stress, this low-shear approach results in intact peptide nanoparticles with good chemical and physical stability. We have demonstrated that this method is applicable to peptides of varying and diverse structural properties, representing a general strategy for high-concentration peptide formulations.

## 2. Results and Discussion

### 2.1. Peptide Nanomilling Approach

Nanosuspensions have been demonstrated as an effective enabled formulation strategy for small molecules [15,16,17,18,19,20,21,22,23]. These formulations generally consist of a suspension of drug nanocrystals and thus exhibit particularly high drug loading in comparison to other engineered nanoparticle delivery systems, such as liposomes and lipid or polymer nanoparticles. Nanosuspensions often consist of nanoparticles of >75% drug load and can often be prepared at a high overall drug concentration in aqueous suspensions (i.e., >100 mg/mL). Due to their small particle size and large surface area, nanosuspension formulations exhibit dramatically increased dissolution rates and potentially saturation solubilities [24], enabling improved absorption in vivo when administered orally. In addition, nanosuspensions can also be administered via parenteral delivery routes as well [25,26,27], including IV and SC delivery [28,29,30,31].

Small-molecule nanosuspensions are often manufactured using top-down wet milling, high-pressure homogenization, or ultrasonication [32]. Bottom-up precipitation methods have been investigated, although these require initial solubility in organic solvents, presenting challenges with purification and isolation as well as stability [25]. Once prepared, the drug nanoparticles themselves are inherently high-energy and unstable, presenting a risk of aggregation. In order to mitigate this risk, small amounts of polymer and/or surfactant excipients are added to stabilize the nanoparticles and prevent aggregation from occurring. This requires the selection of the optimal stabilizer combination, which can be drug-specific and must be identified through empirical screening [25,26,27], although recent computational work has started to elucidate an understanding of the drug stabilizer interactions involved [33].

Nevertheless, these types of manufacturing processes involve high shear forces used to generate a small particle size, which has made these approaches prohibitive for sensitive large molecules such as peptides and proteins. In recent years, milling using acoustic mixing has been discovered as a process that can generate stable nanoparticles using low-shear resonant acoustic waves [14]. Unlike techniques such as ultrasound, resonant acoustic mixing delivers energy to a sample at its resonant frequency, typically only between 58 and 62 Hz [34]. This is an extremely efficient mechanism for mixing while, at the same time, the low frequencies used result in less stressful shear forces to the materials being mixed. As a result, resonant acoustic mixing results in more stable materials compared to earlier high-shear techniques as well as an approach to mill sensitive compounds [14,35]. Thus, we hoped to evaluate the acoustic milling approach as a new, softer technique for preparing stable nanoparticles of sensitive peptides as well as without the use of organic solvents that may disrupt the peptide structure. To demonstrate the generality of this technique, three structurally diverse peptides were studied, insulin (a large peptide hormone), GNE-A (a cystine-knot peptide), and cyclosporine A (CsA, a macrocyclic peptide) (Figure 1). Ideally, these peptide nanoparticles would retain high drug loading and good stability at high concentrations, behaving similarly to small-molecule nanosuspensions.

### 2.2. Insulin Nanosuspensions

Insulin is a large peptide hormone that regulates glucose metabolism in vivo. It consists of 51 amino acid residues in two linear peptide chains with an overall molecular weight of 5.8 kDa. The insulin monomer is the physiologically active agent but is sensitive to instability [36]. Insulin is particularly susceptible to aggregation, forming oligomers such as dimers, tetramers, and hexamers, as well as uncontrolled amyloid fibrils [37]. As a result, historically, insulin has been prepared as a zinc complex, which exists in a more stable hexamer form [38,39]. Thus, most formulation work on insulin has been focused on mitigating aggregation in solution, particularly of the active monomeric form.

There have been a number of reports around the use of nanoparticle approaches for the delivery of insulin, although these are often complex nanocarrier systems such as polymeric and lipid nanoparticles [40]. An initial report from Merisko-Liversidge and coworkers demonstrated nanomilling on an insulin–zinc complex, although this was limited to the more stable and inactive insulin hexamer species rather than the active and more sensitive insulin monomer, which is more desirable for its rapid time of action [41]. Thus, we hoped that the softer resonant acoustic mixing approach could be used to prepare stable nanoparticles of the insulin monomer itself.

In order to determine the appropriate formulation stabilizer composition for the insulin nanoparticles, a nanomilling screen was conducted. As seen in Table 1, the insulin monomer solid was suspended in water and was milled under resonant acoustic mixing conditions in the presence of different commonly used polymer and surfactant stabilizers [33]. The formulations prepared using HPC-SL/SDS and PVP K29-32/SDS resulted in small nanoparticles and were thus selected for scale-up and further study. The average radius of the nanoparticles in both formulations were translatable in the scale-up batch and remained consistent at ~110–130 nm (Table 2).

The chemical and physical stability of the two nanosuspension with HPC-SL/SDS and PVP K29-32/SDS were evaluated. The insulin nanosuspensions were stored at room temperature (RT) as well as at 4 °C and 37 °C for up to 1 month. As can be seen in Figure 2, the particle size of the insulin nanoparticles with PVP K29-32/SDS remained in the accepted nano-scale range for up to a month even when stored at 37 °C. No significant particle size growth was observed after being stored for 28 days at 4 °C or even room temperature. In contrast, the insulin nanosuspensions with HPC-SL/SDS appeared to thicken over time and experienced some particle growth at 4 °C and room temperature as well as significant aggregation at 37 °C.

In order to more closely quantitate the aggregation state and monomer content of the insulin nanosuspensions, the samples were also analyzed via size-exclusion chromatography (SEC). As can be seen in Figure 3, the insulin monomer concentration remains quantitative for both formulations, even after 28 days, for all of the samples with no presence of higher-order oligomers such as dimers, trimers, or hexamers. The chemical stability of the insulin nanosuspension was also investigated using reverse-phase (RP) chromatography (Figure 4). Both of the formulations remained relatively chemically stable with only a small amount of degradation observed for the samples stored under the accelerated condition of 37 °C for 28 days.

These results establish that resonant acoustic milling could be used to identify and prepare discrete nanoparticles of insulin monomers that can remain stable even under accelerated conditions.

### 2.3. GNE-A Nanosuspensions

GNE-A is a cysteine-knot peptide developed at Genentech. It is composed of a 30-residue linear peptide having three internal disulfide bonds with an overall molecular weight of 3.4 kDa. This peptide is highly prone to aggregation via multiple pathways, forming both amorphous non-covalent aggregates as well as oligomers formed from covalent disulfide scrambling [42]. As a result, the ability to overcome these risks and develop a stable high-concentration formulation of GNE-A would be highly valuable.

Similar to the insulin results, an initial nanomilling screen of GNE-A was conducted in order to identify the optimal formulation composition. GNE-A solid was suspended in water and was milled under resonant acoustic mixing conditions in the presence of commonly used polymer and surfactant stabilizers. In this case, stable nanoparticles of GNE-A were observed for a wide range of formulation compositions (Table 3). Two formulations, GNE-A with 25% Pluronic F127 and with 25% Tween 80, were selected for preparation at a larger scale due to their compatibility for parenteral administration. As with the insulin samples, the selected nanosuspension formulations of GNE-A could be successfully prepared at larger scale (Table 4).

As can be seen in Figure 5, nanosuspensions prepared with Tween 80 and Pluronic F127 retained their nanoparticle size after 28 days at room temperature and 4 °C conditions. However, both formulations exhibited a slight increase in observed particle size over time at the higher temperature accelerated condition of 37 °C. In order to quantitate the aggregation state and monomer content of GNE-A more closely, the samples were also analyzed via SEC. As can be seen in Figure 6, the GNE-A monomer concentration remains relatively stable under these conditions, although a slight reduction was seen for the samples stored at 37 °C.

GNE-A is also sensitive to chemical stability liabilities, particularly oxidation. The nanosuspension samples were also analyzed via reverse phase (RP) chromatography using a method developed to quantify the presence of the oxidative degradation product (Figure 7). For these samples, the Tween 80 formulations exhibited increasing amounts of oxidation degradation. This is likely due to the presence of small amounts of residual peroxide products in the Tween 80 material [43]. In contrast, the Pluronic F127 formulation samples remained highly chemically stable with no significant oxidation degradation observed even after 28 days at 37 °C. Thus, the nanosuspension formulations appeared to be highly chemically stable as well.

The stability benefits of the nanosuspension formulations of GNE-A are particularly striking when compared to a corresponding solution formulation prepared at the same concentration. GNE-A can form a high-concentration aqueous solution at pH levels > 7. A solution of GNE-A in 60 mM phosphate buffer at pH 7 was prepared at 100 mg/mL. However, in solution, GNE-A rapidly begins to undergo aggregation and loss of monomer as determined via SEC (Figure 8). In contrast, the nanosuspension formulation at a concentration of 100 mg/mL remained stable with the 100% monomer over 28 days. Thus, a nanoparticle suspension of GNE-A remained physically stable while reducing the tendency of the peptide to directly self-associate and undergo aggregation. Importantly, the active monomer form remained intact under these conditions.

### 2.4. Cyclosporine A Nanosuspension Formulations

Cyclosporine A (CsA) is a macrocyclic peptide consisting of 11 amino acids and a molecular weight of 1.2 kDa. It has low solubility and low permeability, resulting in significant challenges in absorption [27]. Due to aqueous solubility limitations, CsA is typically formulated by being dissolved in a mixture of lipids, surfactants, and cosolvents. Often, high concentrations of lipid-based excipients are required [44,45,46,47]. A current commercial oral formulation of CsA, Sandimmune^®^, consists of an oral solution or liquid-filled capsules with alcohol, corn oil, glycerol, and Labrafil [48]. The corresponding injectable formulation consists of a large amount of Cremophor EL (a polyethoxylated castor oil) and alcohol to achieve the desired solubility. This has resulted in safety issues, such as a potential risk of intolerability at the injection site as well as anaphylaxis reactions [49,50,51,52,53]. Thus, there is a strong desire to develop alternative formulations of CsA without the need for excipients, such as Cremophor EL.

There has been some recent work conducted on aqueous CsA nanosuspensions [54,55]. However, these studies used anti-solvent precipitation, high-pressure homogenization, or high-shear wet milling processes, making it difficult to translate them to other more sensitive peptides. Due to its small size and cyclic structure, CsA acts more like a small molecule and remains stable through these processes. Nevertheless, we wanted to see if resonant acoustic milling could be used as a general process for CsA as well.

As above, for insulin and GNE-A, a nanomilling screen was conducted on CsA in order to identify the optimal formulation composition for forming stable nanoparticles. In the initial screen, only the HPC-SL/SDS and Tween 80 compositions presented stable nanoparticles (Table 5). These two formulations were then prepared at a larger scale for further analysis, which exhibited a slight increase in the average particle size (Table 6).

Dry-state TEM with a negative stain was performed to further characterize the size and morphology of the nanoparticles in the nanosuspension. The CsA nanosuspension with HPC-SL/SDS was prepared at a concentration of 100 mg/mL and stored at 4 °C for 1 month before the sample was imaged. An aliquot of the formulation was taken and diluted to 1 mg/mL with deionized water for imaging. The TEM image (Figure 9) suggests that the particle size of the CsA nanoparticles is <200 nm in diameter.

The longer-term stability of these formulations was then investigated (Figure 10). With respect to the average particle size, both formulations appeared relatively stable for up to 1 month at 4 °C as well as at room temperature. However, some larger aggregation was observed over time in the samples stored at 37 °C, potentially leading to the settlement of the larger particles (explaining the smaller average particle sizes observed in the samples after 1 month). While this suggested that cold storage could be used to mitigate physical stability risks, we hoped to identify more stable nanosuspension formulations.

The challenges in the nanomilling process, particularly during scale-up, were likely due in part to the high lipophilicity of CsA, resulting in wettability issues. In order to optimize the process, higher concentrations of surfactant stabilizers were used to aid in the wetting and milling performance. As can be seen in Figure 11, when increasing the ratios of SDS, a surfactant stabilizer and wetting agent added to CsA, the milling efficiency increased, and the resulting particle sizes continued to decrease. The stability of the CsA nanosuspension with 25% SDS (wt% to CsA) was investigated (Figure 12). Although some variability was observed, no significant increase in aggregation was observed over 28 days. The concentration of the monomer was characterized via SEC, which showed no change over 28 days even at conditions up to 37 °C (Figure 13). Similarly, no chemical degradation or growth of degradants was observed via RP chromatography (Figure 14).

In addition, while the average particle size decreased as the SDS concentration increased, the solubility of the formulation was also measured to see if there was an additional effect with the addition of a higher quantity of the surfactant. CsA is practically insoluble in water (0.04 mg/mL) [56]. However, increasing concentrations of SDS dramatically increase the solubility of CsA in a proportional manner (Table 7).

The viscosities of the CsA nanosuspension formulation with 25% SDS (wt% to CsA) were measured and compared to the commercial Sandimmune^®^ solution to determine the feasibility of processing and injection at high concentrations. The CsA nanosuspension was prepared at 100 mg/mL and diluted to 25 mg/mL and 5 mg/mL with water. The commercial Sandimmune was purchased as a 50 mg/mL lipid-based formulation and was diluted to 25 mg/mL and 5 mg/mL with saline based on the package injection instructions. The two concentrations were chosen as the low and high doses for subcutaneous injection administration. The CsA nanosuspension exhibited a low viscosity of 2.1 Pa·s, even at a high concentration of 100 mg/mL, whereas the commercial Sandimmune^®^ formulation only showed a similar viscosity at a much lower concentration of 5 mg/mL. Moreover, the Sandimmune^®^ formulation showed significantly higher viscosities, even at 25 mg/mL, with a value of 297 Pa·s (Table 8). Interestingly, the viscosity at 25 mg/mL was observed to be higher than the viscosity at 50 mg/mL in the commercial Sandimmune formulation. This may be due to the interaction of differing ratios of the lipid and aqueous phases at this shear range. Nevertheless, the viscosities of the CsA nanosuspension at all concentrations were dramatically lower and within the acceptable range with no injection concerns.

The injection force and syringeability of both the CsA nanosuspension formulation and the commercial Sandimmune^®^ solution formulation was also evaluated at a concentration of 25 mg/mL. The purpose of this study was to mimic the scenario of a subcutaneous injection at a high dose by using the same equipment. There was a strong back pressure when drawing up the Sandimmune^®^ solution formulation with a 25 G needle, but the formulation could be pushed out without any resistance. Therefore, the formulation was initially extracted using a larger 18 G needle and then pushed out after switching back to a 25 G needle. Being consistent with the higher viscosity values, the Sandimmune^®^ lipid-based solution formulation exhibited a significantly higher injection force compared to the aqueous nanosuspension formulation (Figure 15). Over a 5 s injection, the maximum forces experienced by the Sandimmune^®^ solution formulation and the nanosuspension formulation were 26.97 N and 4.33 N, respectively.

The in vivo pharmacokinetics of CsA prepared as either the lipid-based Sandimmune^®^ solution formulation or the nanosuspension prepared via resonant acoustic milling were evaluated following a single subcutaneous injection at 10 mg/kg (5 mg/mL formulation concentration) or 50 mg/kg (25 mg/mL formulation concentration), with a dosing volume of 2 mL/kg. As previously stated, the needle switching strategy was implemented on the high-dosed Sandimmune^®^ formulation due to its high injection force, whereas there was no injection concern for the aqueous nanosuspension formulations. Over a period of 24 h, all of the formulations exhibited a sustained release with steady plasma concentrations with no sign of decreasing (Figure 16). The half-life of CsA in rats has been reported to range from 7.4–12 h [57], which indicates that continuous absorption from the injection site occurred over the study time period. This suggests that a longer study may be needed to identify the total AUC of the formulations. The nanosuspension at both 10 mg/kg and 50 mg/kg also exhibited a lower Cmax and absorption compared to the commercial Sandimmune^®^ formulations (Table 9). The lower steady state plasma concentrations of the nanosuspensions are likely due to the solubility-limited absorption of CsA in the formulations. As measured in Table 6, the solubility of CsA in the 25% SDS (wt% to CsA) nanosuspension is 3.848 mg/mL, while the Sandimmune^®^ formulations consisted of fully dissolved peptides at 5 and 25 mg/mL concentrations, which is likely a major contributing factor for the lower concentrations of absorbed peptides at a steady state. At the 50 mpk dose level, the Sandimmune^®^ formulation resulted in an observed AUC of 59.2 h*μmol/L, while the CsA nanosuspension presented an observed AUC of 7.49 h*μmol/L, which is similar in magnitude to the difference in solubility between both formulations. Likely, longer time points would continue to capture further absorption. Nevertheless, the absorption of the nanosuspensions remains constant over the time period tested, and the plasma concentrations achieved provide sufficient coverage over the target concentrations for the treatment of patients with liver transplants without the need for harsh excipients [58].

### 2.5. Comparison with Alternative High-Energy Milling Approaches

These results demonstrate that resonant acoustic milling can be used to prepare stable nanosuspension formulations comprising a wide variety of peptides with diverse sizes and structures. These peptide nanoparticles exhibit improved physical and chemical stability as well as benefits in formulation development and in vivo performance. In order to demonstrate that this is due to the unique nature of the resonant acoustic mixing effect, we subjected the same peptide materials to more traditional top-down nanomilling approaches as a comparison. In this case, the preparation of the peptide nanosuspensions was attempted using an ultrasonic probe [59,60,61,62]. Unlike resonant acoustic mixing, an ultrasonic probe supplies high-intensity, high-frequency waves (often above 20 kHz) to a sample in order to induce homogenization and reduce particle size [63]. All three peptides were subjected to ultrasonication at 5 and 10 min (Table 10). In these studies, after ultrasonication, the resulting suspensions consisted of heterogeneous particle sizes that were larger than those obtained from resonant acoustic mixing (Figure 17). Furthermore, the suspensions visibly settled over time, showing that they were not small enough or stable enough to remain in suspension (Figure 18). When the peptides were sonified beyond 10 min in an attempt to further reduce the particle size of the samples, they noticeably appeared to turn into gel due to high-intensity ultrasonication.

## 3. Materials and Methods

### 3.1. Materials

Compounds, reagents, and solvents were obtained from commercial sources and used as received, unless otherwise noted. Insulin (recombinant human) was obtained from Millipore Sigma Cat# 91077C-1G. GNE-A is a cysteine-knot peptide and was obtained from Genentech Research Laboratories, South San Francisco, CA, USA. Cyclosporine A was obtained from Toronto Research Chemicals (North York, ON, Canada). Sandimmune^®^ Injection (cyclosporine, USP) 50 mg/mL was obtained from McKesson Medical (Richmond, VA, USA). Formic acid was obtained from Alfa Aesar (Haverhill, MA, USA). Acetonitrile was obtained from VWR (West Chester, PA, USA). Trifluoroacetic acid was obtained from J.T. Baker (Phillipsburg, NJ, USA). Ammonium formate was obtained from Sigma-Aldrich, St. Louis, MO, USA. Sodium Dodecyl sulfate (SDS) was obtained from Spectrum Chemical (New Brunswick, NJ, USA). Plasdone (PVP) K29-32 was obtained from Acros Organics (Geel, Belgium). Pluronic F127 was obtained from Sigma. Tween 80 was obtained from Sigma. Hydroxypropyl Cellulose (HPC)-SL was obtained from Alfa Aesar. Loadings and concentrations are reported as weight percentages (wt%), unless otherwise noted.

### 3.2. Nanosuspension Screening Using Resonant Acoustic Milling

A UV-Star clear, flat-bottom 96-well plate was used as a high-throughput mixing container, as described previously [14]. Each well plate was charged with 500 μm YTZ grinding media from Tosoh (800 mg, 175 μL by volume) (Tosoh USA, Inc., Grove City, OH, USA), 2 mg of peptide powder (1.3% drug loading), and 148 μL of an aqueous excipient solution. The concentrations of the polymer and/or surfactant excipients varied between 0.006% and 1.95% within each well plate. Each well plate was sealed with a Thermo Fisher Scientific ALPS 50 V Manual Heat Sealer (Thermo Fisher Scientific Inc., Waltham, MA, USA). Each sealed well plate was then placed on a Resodyn LabRAM II Resonant Acoustic mixer (Resodyn Acoustic Mixers, Butte, MT, USA) and milled at 50 G acceleration for 2 h.

### 3.3. Nanosuspension Scale-Up

A 4 mL clear glass vial was used as a scale-up container. The vial was charged with 9.12 g of 500 μm YTZ grinding media from Tosoh, 175 mg of peptide powder (10% drug loading), and 1.575 mL of an aqueous excipient solution. The vial was then placed on a Resodyn LabRAM II Resonant Acoustic mixer and milled at 50 G acceleration for 2 h. The resulting nanosuspension was recovered by using a syringe equipped with an 18 G needle.

### 3.4. Analytical Characterization Methods

#### 3.4.1. Particle Size Analysis Using Dynamic Light Scattering

After milling, a 5 μL aliquot of the nanosuspension sample was taken and diluted in 995 μL of D.I. water for analysis using a Wyatt DynaProTM Plate Reader II (Waters Corporation, Santa Barbara, CA, USA) dynamic light scattering instrument. A 30 μL aliquot of the diluted suspension was dispensed into a Corning^®^ low-volume black polystyrene 384-well plate for analysis. The particle size of each sample was reported as an average of 10 acquisitions with an acquisition time of 5 s at 25 °C. Autocorrelation curves were fitted using either the cumulants or regularization method, and the average particle radius and diameters D50 and D90 were obtained and reported along with their standard deviations. The normalized polydispersity (%Pd) was calculated as the polydispersity divided by the estimated hydrodynamic radius from the cumulants fit of the autocorrelation function multiplied by 100.

#### 3.4.2. Stability Analysis Using HPLC

Aliquots of the nanosuspension formulation samples were added to Eppendorf tubes and stored at room temperature, 4 °C, and 37 °C. Samples were taken at time points corresponding to day 0, 14; at each time point, the physical and chemical stability of the formulations were investigated via HPLC. The injected sample dissolved at a concentration of 0.5 mg/mL with 200-fold dilution in 30/70 acetonitrile/water for GNE-A and cyclosporin A or in 0.01 N HCl for insulin.

An Agilent 1290 Infinity II (Agilent Technologies, Santa Clara, CA, USA) was used for chromatographic analysis. Data were acquired and processed using Empower 3 software (Waters, Milford, MA, USA). For all three peptides (GNE-A, cyclosporin A, and insulin), separations were performed in an Xbridge BEH125 SEC column (3.5 mm, 7.8 × 150 mm) for physical stability analysis. Isocratic method was applied in which 80% of the 5 mM phosphate buffer at pH 7 in acetonitrile was used as the mobile phase. The injection volume was 8 μL and the column temperature was 30 °C. The detection wavelength was 214 nm, and the flow rate was 0.5 mL per minute.

For chemical stability, GNE-A and insulin were analyzed by using a Halo Peptide ES-CN column (2.7 mm, 3.0 × 150 mm). In total, 0.1% formic acid in 10 mM ammonium formate (pH 3.2) was used as mobile phase A, and 0.1% formic acid in 80/20 acetonitrile/10 mM ammonium formate was used as mobile phase B. The injection volume was 5 μL, and the column temperature was 35 °C. The detection wavelength was 214 nm, and the flow rate was 0.3 mL per minute. On the other hand, cyclosporin A was analyzed by using a BEH C18 column (1.7 mm, 2.1 × 150 mm). Moreover, 0.1% trifluoroacetic acid in water was used as mobile phase A, and 0.1% trifluoroacetic acid in acetonitrile was used as mobile phase B. The injection volume was 5 μL, and the column temperature was 45 °C. The detection wavelength was 214 nm, and the flow rate was 0.5 mL per minute.

#### 3.4.3. Dry-State Transmission Electron Microscopy

Prior to usage, all buffer and stain aliquots underwent filtration using 0.22 μM spin filters. FCF300-CU grids were glow discharged for 7 s at 15 mA using the CEMRC GlowQube. Subsequently, a 3 μL sample was applied to the grid and left for 1 min. The grids were then washed with 2 × 20 μL drops of a dilution buffer, followed by a 1 × 20 μL drop of a 1% (*w*/*v*) uranyl acetate solution as a stain. The grids were allowed to float on the stain for 1 min before being left to dry. All TEM images were acquired using the Talos L120C microscope (Thermo Fisher Scientific Inc., Waltham, MA, USA), operating at 120 kV with a spot size of 3. The micrographs were recorded on a 4 K × 4 K Thermo Fisher Scientific Ceta CMOS Camera.

#### 3.4.4. Viscosity Measurements

The viscosity was measured using a TA instruments HR-30 Discovery Hybrid Rheometer (Waters, New Castle, DE, USA), equipped with a 20 mm stainless steel 1° angle cone. All samples were allowed to equilibrate at 25 °C prior to testing and a solvent trap was used to prevent solvent evaporation. The sample volume for each sample was 40 μL. The sample viscosity was measured every 15 s for 2.5 min at a constant shear rate of 1000/s. The viscosity (mPas or cP) was calculated via shear stress (Pa) divided by shear rate (1/s).

#### 3.4.5. Injection Force Measurements

The injection force was measured using an Instron Materials Testing System (Model 5542; Norwood, MA, USA) with a 100 N load cell, a syringe holder fixture, a syringe plunger compression plate, and a glass vial to collect the expelled solution.

The samples (25 mg/mL) were prepared by attaching a 25 G BD PrecisionGlide Needle (P/N 305122) to a BD 1 mL Luer-Lok syringe (P/N 309628) and by extracting approximately 0.5 mL of solution into the syringe. The syringe and needle were primed to 0.3 mL, removing any air bubbles in the syringe. The syringe was placed into the syringe holder, and the Instron crosshead was lowered to contact the syringe plunger rod. The program was initiated, displacing the Instron crosshead 17.299 mm at 192 mm/min speed, while recording the associated injection force.

### 3.5. Cyclosporine A In Vivo Pharmacokinetic Studies

The pharmacokinetics of cyclosporine A (CsA) prepared in either the commercial Sandimmune^®^ formulation consisting of (*w*/*w*) 32.9 Ethanol/67.1 Cremophor EL or a nanosuspension formulation were evaluated following a single subcutaneous dose (SC) of 10 or 50 mg/kg to male Sprague Dawley rats with a dose volume of 2 mL/kg. Male SD rats (6–9 weeks old), ranging from 237 to 251 g, and obtained from Charles River Laboratories (Hollister, CA, USA) were used in this study, with 4 rats per dose group. Animals were not fasted before subcutaneous dose administration. Blood samples (approximately 0.15 mL) were collected from each animal via jugular vein into tubes containing K2EDTA at 0.083, 0.25, 0.5, 1, 2, 4, 8, and 24 h after dose administration. Blood was centrifuged at 12,851× *g* for 5 min to harvest plasma. Compound concentration in each plasma sample was determined with a non-validated LC-MS/MS assay at Genentech Inc. (South San Francisco, CA, USA). The lower limit of quantitation (LLOQ) of CsA in plasma was 9.14 ng/mL or 0.00760 μM. Mean CsA concentrations that measured plasma were used to construct a semi logarithmic plasma concentration–time curve. PK analysis was performed using nominal time, non-compartmental analysis, linear-up log-down calculation, and the extravascular input model (model type: Plasma 200–202), Phoenix™ WinNonlin^®^, version 8.3 (Certara L.P., Princeton, NJ, USA).

## 4. Conclusions

The ability to prepare stable peptide formulations at high concentrations would provide significant value for patients as it would enable self-administered subcutaneous delivery. Peptide nanoparticles are one approach to overcome limitations to solubility, dissolution rate, viscosity, and stability, but traditional techniques for preparing nanosuspension formulations have not been generally amenable to peptides due their high shear. In contrast, the results reported here indicate that low-shear resonant acoustic mixing can be used as a method for preparing stable peptide nanoparticles. This approach appears to be highly general and was demonstrated to be effective across a wide range of different peptide structures and molecular weights. These peptide nanosuspensions exhibit improved chemical and physical stability compared to corresponding solution formulations, particularly at high concentrations. In addition, cyclosporine A nanosuspensions were dosed in vivo and exhibit a prolonged and sustained release rate, with the benefit of not requiring high concentrations of lipids and surfactants. As a result, this is a general strategy that should enable the formulation of peptides at high concentrations for a number of applications.

## Figures and Tables

**Figure 1 pharmaceuticals-17-00015-f001:**
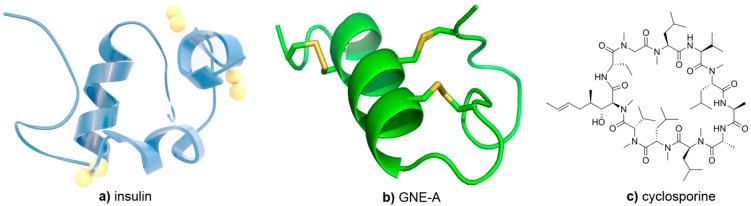
Structures of (**a**) insulin, (**b**) GNE-A, and (**c**) cyclosporine.

**Figure 2 pharmaceuticals-17-00015-f002:**
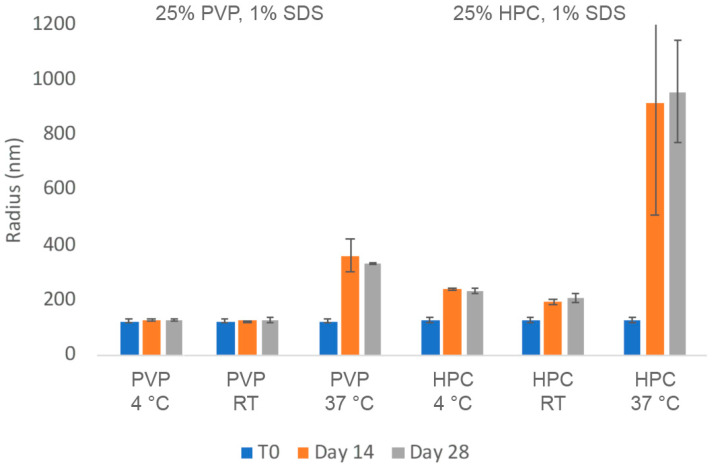
Average particle size of insulin nanosuspension in two selected formulations (wt% to insulin): (1) 25% PVP K29-32, 1% SDS, and (2) 25% HPC-SL, 1% SDS, prepared using resonant acoustic milling measured via dynamic light scattering.

**Figure 3 pharmaceuticals-17-00015-f003:**
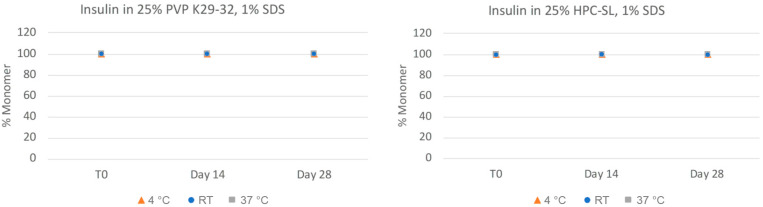
Physical stability of insulin nanosuspensions in two selected formulations (wt% to insulin): (1) 25% PVP K29-32, 1% SDS, and (2) 25% HPC-SL, 1% SDS, prepared using resonant acoustic milling measured via size-exclusion chromatography.

**Figure 4 pharmaceuticals-17-00015-f004:**
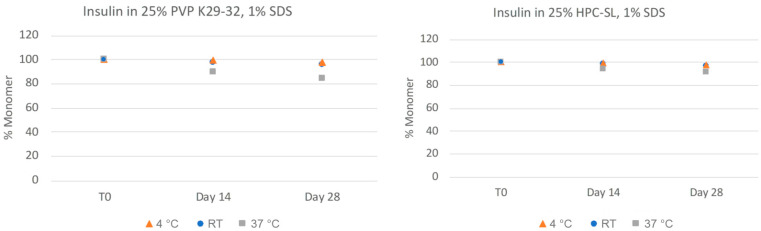
Chemical stability of insulin nanosuspensions in two selected formulations (wt% to insulin): (1) 25% PVP K29-32, 1% SDS, and (2) 25% HPC-SL, 1% SDS, prepared using resonant acoustic milling measured via reverse-phase chromatography.

**Figure 5 pharmaceuticals-17-00015-f005:**
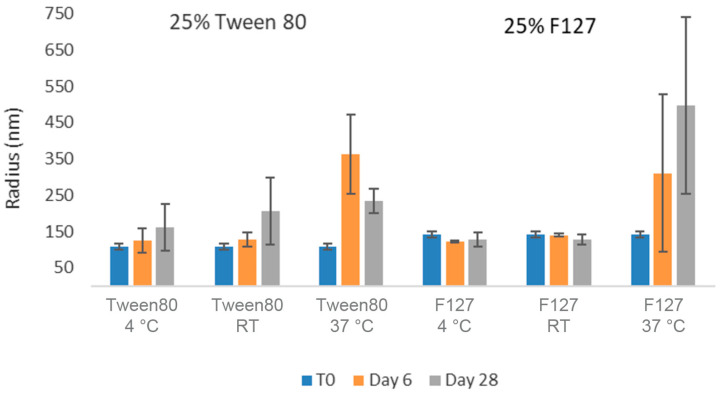
Average particle size of GNE-A nanosuspensions in two selected formulations (wt% to GNE-A): (1) 25% Tween80 and (2) 25% Pluronic F127 using acoustic milling measured via dynamic light scattering.

**Figure 6 pharmaceuticals-17-00015-f006:**
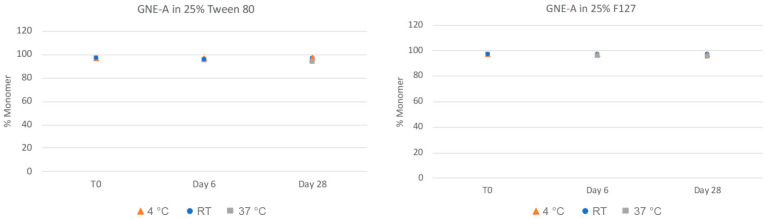
Physical stability of GNE-A nanosuspensions in two selected formulations (wt% to GNE-A): (1) 25% Tween80 and (2) 25% Pluronic F127 prepared using resonant acoustic milling measured via size-exclusion chromatography.

**Figure 7 pharmaceuticals-17-00015-f007:**
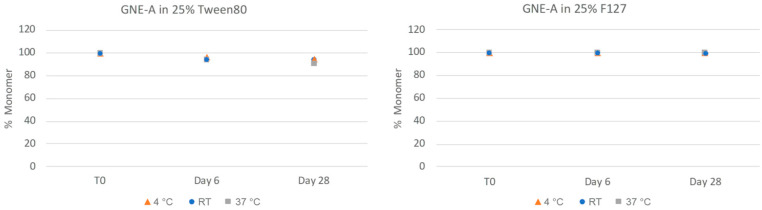
Chemical stability of GNE-A nanosuspensions in two selected formulations (wt% to GNE-A): (1) 25% Tween80 and (2) 25% Pluronic F127 prepared using resonant acoustic milling measured by reverse-phase chromatography.

**Figure 8 pharmaceuticals-17-00015-f008:**
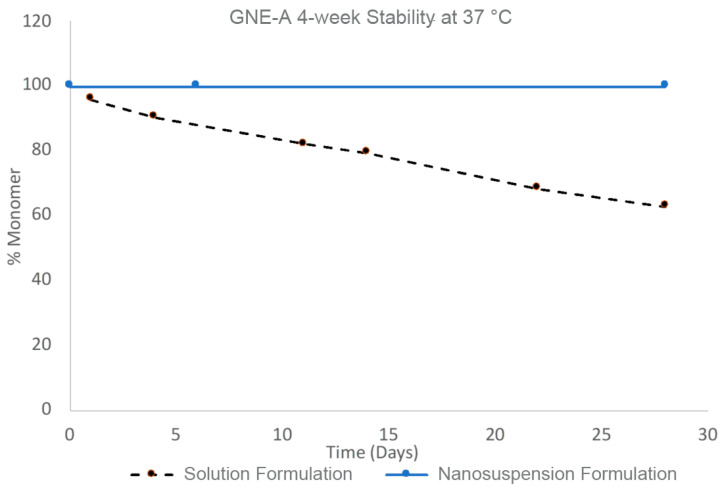
Four-week stability of GNE-A nanosuspension with Pluronic F127 and solution formulations at 100 mg/mL concentration at 37 °C analyzed via size-exclusion chromatography.

**Figure 9 pharmaceuticals-17-00015-f009:**
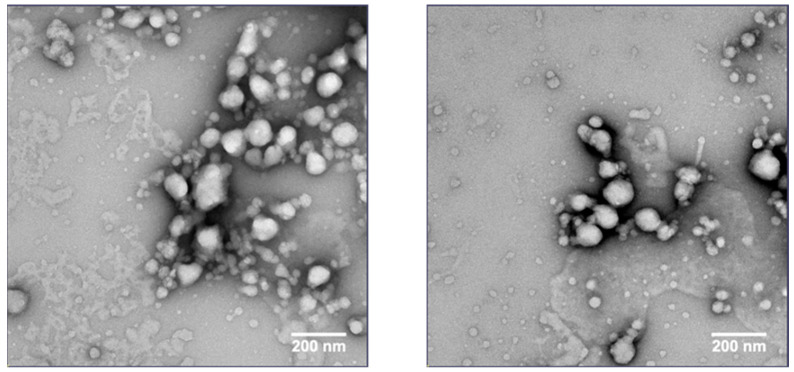
TEM images of CsA nanosuspension with 25% HPC-SL, 1% SDS (wt% to CsA).

**Figure 10 pharmaceuticals-17-00015-f010:**
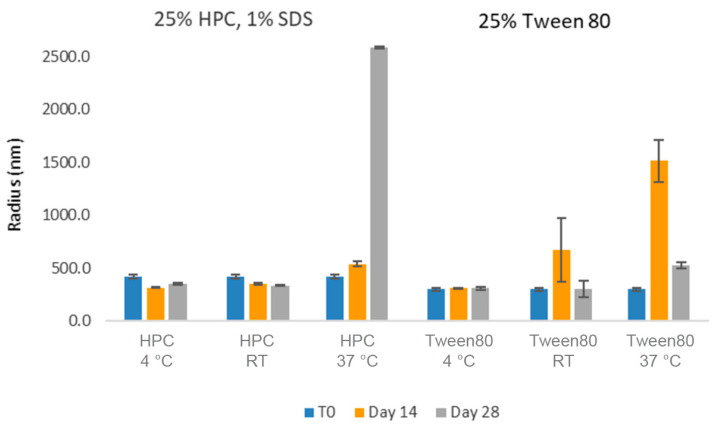
Average particle size of CsA nanosuspensions in two selected formulations (wt% to CsA): (1) 25% HPC-SL, 1% SDS, and (2) 25% Tween 80 using acoustic milling measured by dynamic light scattering.

**Figure 11 pharmaceuticals-17-00015-f011:**
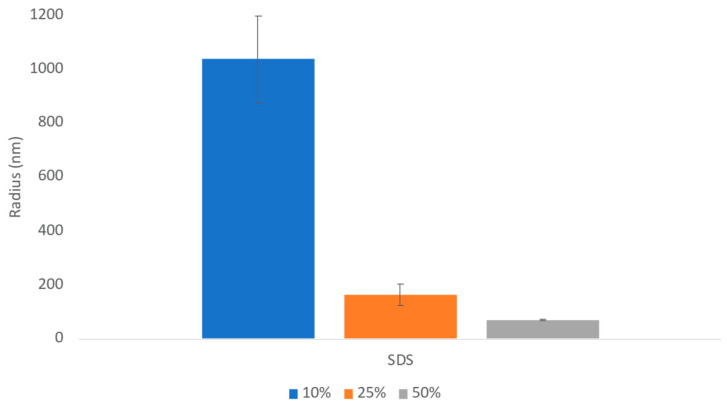
Average particle size of CsA nanosuspensions with increasing ratios of SDS at 10 mg/mL concentration measured via dynamic light scattering.

**Figure 12 pharmaceuticals-17-00015-f012:**
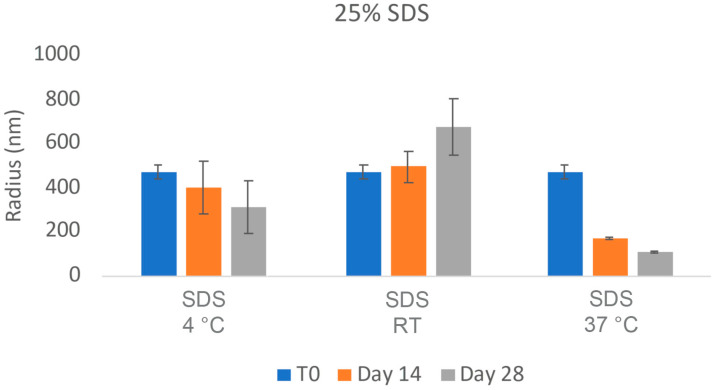
Average particle size of CsA nanosuspensions with 25% SDS (wt% to CsA) over 24 h at 10 mg/mL concentration measured via dynamic light scattering.

**Figure 13 pharmaceuticals-17-00015-f013:**
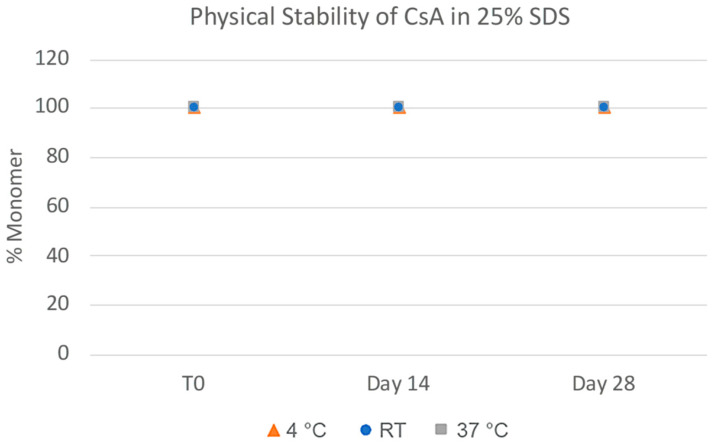
Physical stability of CsA nanosuspensions with 25% SDS (wt% to CsA) prepared using resonant acoustic milling measured via size-exclusion chromatography.

**Figure 14 pharmaceuticals-17-00015-f014:**
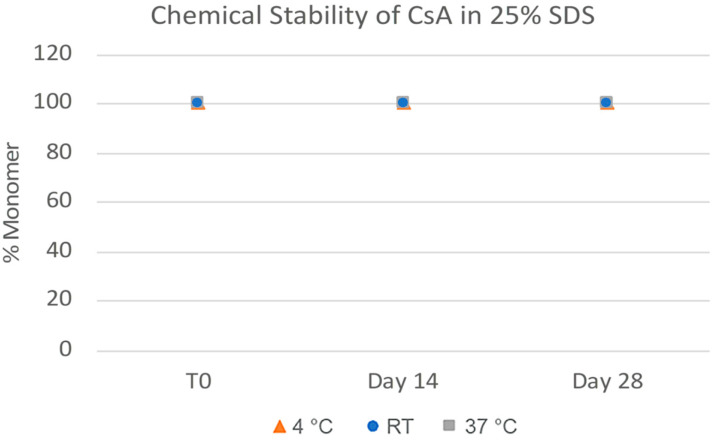
Chemical stability of CsA nanosuspensions with 25% SDS (wt% to CsA) prepared using resonant acoustic milling measured via reverse-phase chromatography.

**Figure 15 pharmaceuticals-17-00015-f015:**
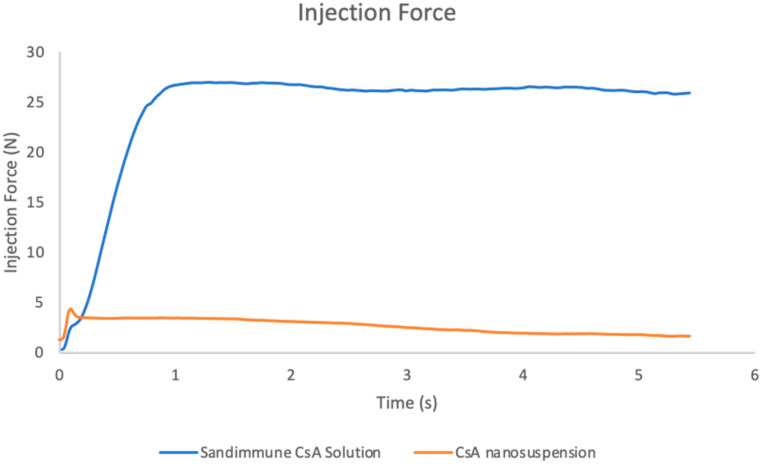
Injection forces of Sandimmune^®^ lipid-based formulation and aqueous nanosuspension of CsA by using a BD 1 mL syringe attached with a 25 G needle. The maximum forces of Sandimmune^®^ formulation and CsA nanosuspension were 26.97 N and 4.33 N, respectively.

**Figure 16 pharmaceuticals-17-00015-f016:**
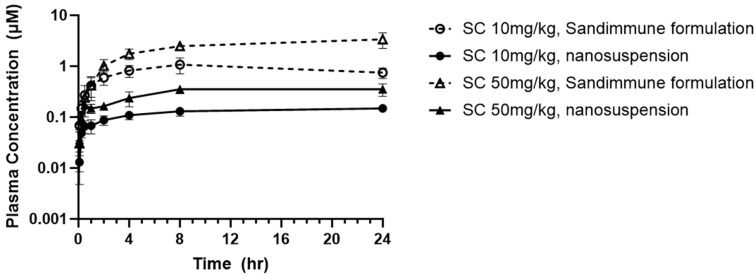
PK profile of commercial Sandimmune^®^ formulation and aqueous nanosuspension formulation of CsA.

**Figure 17 pharmaceuticals-17-00015-f017:**
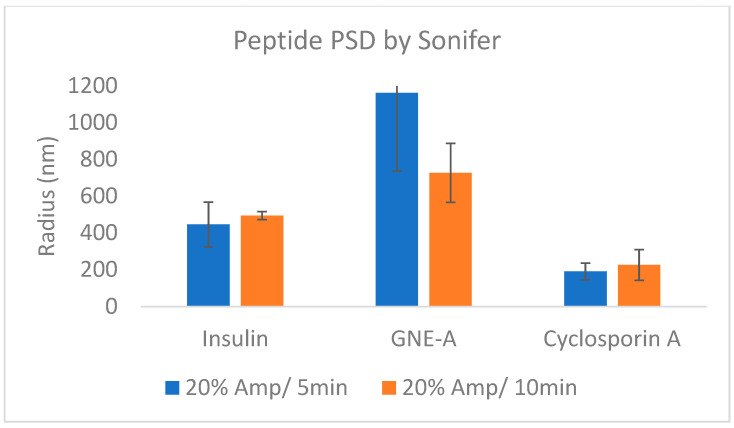
Average particle sizes of peptide formulations prepared using ultrasonication.

**Figure 18 pharmaceuticals-17-00015-f018:**
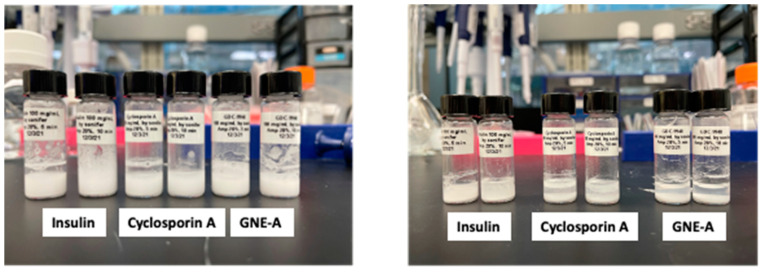
Peptide formulations milled for 5 and 10 min using ultrasonication exhibiting settling behavior from the initial time point (**left**) until day 3 (**right**).

**Table 1 pharmaceuticals-17-00015-t001:** Nanomilling screen conducted on insulin monomer milled at 100 mg/mL concentration.

Peptide	Formulation (wt% to Insulin)	Average Radius (nm)	% Pd
Insulin Monomer	25% HPC-SL, 1% SDS	130 ± 4	28.3
	25% PVP K29-32, 1% SDS	112 ± 2	23.7
	25% Pluronic F127	419 ± 24	14.6
	25% Tween80	903 ± 124	Multimodal

**Table 2 pharmaceuticals-17-00015-t002:** Particle size of the insulin nanosuspensions prepared at larger scale at 100 mg/mL concentration.

Peptide	Formulation (wt% to Insulin)	Average Radius (nm)	% Pd
Insulin Monomer	25% HPC-SL, 1% SDS	135 ± 14	33.6
	25% PVP K29-32, 1% SDS	121 ± 5	19.9

**Table 3 pharmaceuticals-17-00015-t003:** Nanomilling screen conducted on cysteine-knot peptide GNE-A milled at 100 mg/mL concentration.

Peptide	Formulation (wt% to GNE-A)	Average Radius (nm)	% Pd
GNE-A	25% HPC-SL, 1% SDS	142 ± 3	14
	25% PVP K29-32, 1% SDS	161 ± 37	Multimodal
	25% Pluronic F127	134 ± 5	16.4
	25% Tween 80	190 ± 8	21.4

**Table 4 pharmaceuticals-17-00015-t004:** Particle size of the scale-up nanomilling on cystine-knot peptide GNE-A at 100 mg/mL concentration.

Peptide	Formulation (wt% to GNE-A)	Average Radius (nm)	% Pd
GNE-A	25% Pluronic F127	137 ± 4	27.3
	25% Tween 80	104 ± 4	19.1

**Table 5 pharmaceuticals-17-00015-t005:** Nanomilling screen conducted on macrocyclic peptide CsA milled at 100 mg/mL concentration.

Peptide	Formulation (wt% to CsA)	Average Radius (nm)	% Pd
Cyclosporine A	25% HPC-SL, 1% SDS	307 ± 20	49.1
	25% PVP K29-32, 1% SDS	1177 ± 295	Multimodal
	25% Pluronic F127	480 ± 77	Multimodal
	25% Tween80	193 ± 7	21.8

**Table 6 pharmaceuticals-17-00015-t006:** Particle size of the scale-up nanomilling on CsA at 100 mg/mL concentration.

Peptide	Formulation (wt% to CsA)	Average Radius (nm)	% Pd
Cyclosporine A	25% HPC-SL, 1% SDS	400 ± 32	31.9
	25% Tween80	316 ± 14	26.9

**Table 7 pharmaceuticals-17-00015-t007:** Solubility of CsA nanosuspensions with different ratios of SDS (wt% to CsA).

Solubility (mg/mL)	0% SDS	10% SDS	25% SDS	50% SDS
Cyclosporine A	0.038	1.047	3.848	8.599

**Table 8 pharmaceuticals-17-00015-t008:** Viscosities of CsA nanosuspension (SDS at 25 wt% to CsA) and Sandimmune^®^ solution formulation.

Viscosity (Pa·s/CsA)	5 mg/mL	25 mg/mL	50 mg/mL	100 mg/mL
CsA Nanosuspension	0.76	0.86	n/a	2.1
Sandimmune^®^ CsA Solution	2.1	297	92	n/a

**Table 9 pharmaceuticals-17-00015-t009:** Pharmacokinetic parameters of commercial Sandimmune^®^ and nanosuspension formulations dosed via subcutaneous injection in rat at 10 and 50 mg/kg.

Formulation	Dose (mg/kg)	AUClast (h*μmol/L)	Cmax (μmol/L)	Tmax (h)
Sadimmune^®^ (CsA Solution)	10	20.3	1.13	12
Sandimmune^®^ (CsA Solution)	50	59.2	3.38	24
Nanosuspension (CsA)	10	3.04	0.149	24
Nanosuspension (CsA)	50	7.49	0.409	11

**Table 10 pharmaceuticals-17-00015-t010:** Nanosuspension behavior after preparation using ultrasonication.

Peptide Nanosuspension (wt% to Peptide)	Amp 20%/5 min	Amp 20%/10 min
CsA in 25% HPC-SL, 1% SDS	particle settled at bottom	particle settled at bottom
GNE-A in 25% F127	particle settled at bottom	particle settled at bottom
Insulin in 25% PVP K29-32, 1% SDS	homogenous	homogenous

## Data Availability

Data is contained within the article.

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
