# Peer review of "A New Approach for Preparing Stable High-Concentration Peptide Nanoparticle Formulations"

_pharmaceuticals, 2023, doi:10.3390/ph17010015_

Round 1

Reviewer 1 Report

Comments and Suggestions for Authors

The manuscript presents the acoustic milling approach as a new, softer technique for preparing stable nanoparticles of sensitive peptides without using organic solvents that may disrupt the peptide structure. The authors demonstrated the generality of this technique on three structurally diverse peptides like cyclosporine A (CsA, a macrocyclic peptide), GNE-A (a cystineknot peptide), and insulin (a large peptide hormone). The rationale for the study is very well documented with a prepared introduction. The experiment is well planned and the carried out steps are well described.

The manuscript should be supplemented according to the following points:

1. The studies conducted on the analysis of nanoparticle size and physical and chemical stability did not include statistical analysis. When evaluating these parameters, it is necessary to include the number of experiments, SD, RSD, confidence interval, etc. in order to properly evaluate the precision and accuracy. Please complete this by including appropriate tables in SI and in the graphs included in the paper.

2. The resolution of the chemical and physical stability plots is of poor quality. The font in the title is too large, the line diameter is too large, no confidence intervals are marked, etc. I would like to ask you to improve the quality of the presented results.

3. Please explain what methods and solvents were used to analyze each compound by HPLC, as it is unclear.

In line 133 “Isocratic method was applied in which 80% of the 5 mM Phosphate buffer at pH 7 in acetonitrile was used as the mobile phase.”

In line 138 “0.1% formic acid in 10 mM ammonium formate (pH 3.2) was used as 138 mobile phase A and 0.1% formic acid in 80/20 acetonitrile/10mM ammonium formate was 139 used as mobile phase B.”

In line 143 “0.1% trifluoroacetic acid 143 in water was used as mobile phase A and 0.1% trifluoroacetic acid in acetonitrile was used 144 as mobile phase B.”

4. Why was stability tested only for 28 days? Will the drug product prepared in this form be valid for administration after 28 days? It would be worthwhile to extend the stability studies beyond 28 days. Not necessarily in this study. Are there any plans to do so?

5. Please justify the choice of polymers or stabilizers used in the formulation.

Author Response

We thank the reviewer for their thoughtful comments and suggestions.  Please see our responses and revisions regarding each point below:

The manuscript should be supplemented according to the following points:

  1. The studies conducted on the analysis of nanoparticle size and physical and chemical stability did not include statistical analysis. When evaluating these parameters, it is necessary to include the number of experiments, SD, RSD, confidence interval, etc. in order to properly evaluate the precision and accuracy. Please complete this by including appropriate tables in SI and in the graphs included in the paper.

The analytical characterization methods and treatment of data are included in Section 3 Materials and Methods.  For nanoparticle size, dynamic light scattering was used and the average particle size was reported as an average of 10 acquisitions with an acquisition time of 5 sec at 25 °C.  The data was fit using the cumulants model.  Per the reviewer’s suggestion, we have included the standard deviation of the 10 acquisitions in the corresponding tables in the manuscript to provide precision.  For the stability data, due to material availability, each time point was taken as an n=1, although we believe that the stability trends over multiple time points provide a consistent reflection of stability across all of the samples. 

  1. The resolution of the chemical and physical stability plots is of poor quality. The font in the title is too large, the line diameter is too large, no confidence intervals are marked, etc. I would like to ask you to improve the quality of the presented results.

We thank the reviewer for their comment and have updated the resolution and quality of the figures.

  1. Please explain what methods and solvents were used to analyze each compound by HPLC, as it is unclear.

In line 133 “Isocratic method was applied in which 80% of the 5 mM Phosphate buffer at pH 7 in acetonitrile was used as the mobile phase.”

In line 138 “0.1% formic acid in 10 mM ammonium formate (pH 3.2) was used as 138 mobile phase A and 0.1% formic acid in 80/20 acetonitrile/10mM ammonium formate was 139 used as mobile phase B.”

In line 143 “0.1% trifluoroacetic acid 143 in water was used as mobile phase A and 0.1% trifluoroacetic acid in acetonitrile was used 144 as mobile phase B.”

We thank the reviewer for bringing this to our attention.  We have revised the methods section to more clearly specify which method was used for which peptide as well as for which purpose (physical stability by SEC or chemical stability by reverse phase chromatography).

  1. Why was stability tested only for 28 days? Will the drug product prepared in this form be valid for administration after 28 days? It would be worthwhile to extend the stability studies beyond 28 days. Not necessarily in this study. Are there any plans to do so?

 In these initial studies, chemical and physical stability testing was conducted over 28 days.  This was to provide sufficient preliminary data to demonstrate differentiation of the behavior of these peptide formulations from each other (e.g., Figure 8).  In addition, both standard conditions (refrigerated at 4 °C) as well as accelerated conditions at higher temperatures (37 °C) were conducted.  The stability results under these accelerated conditions can be extrapolated for longer stability at lower temperatures.  Nevertheless, the reviewer is correct that for a commercial product, more intensive and longer stability studies would be required to qualify these drug products.

  1. Please justify the choice of polymers or stabilizers used in the formulation.

The polymer and surfactant stabilizers used in this study were derived from similar formulation compositions used for small molecule nanosuspension formulations.  We have included a reference to a paper from Ferrer et al. that describes the interactions of these stabilizers towards nanoparticle surfaces. 

Reviewer 2 Report

Comments and Suggestions for Authors

The authors investigate the preparation of stable high concentration peptide nanoparticles. The subject involves several considerations to ensure their stability and prevent aggregation. Key factors that are considered are the peptide design, selection of the solvent, pH optimization, concentration optimization, temperature dependence, mixing techniques, prevention of contamination, stabilizing agents, and characterization by physicochemical methods. The particular novelty of the paper is the use of low shear resonant acoustic mixing for peptide nanoparticle preparation.

I have a few suggestions for improvement:

1.       The polydispersity data reported in Tables 1 and 2 appear to be very high. Even in some cases, a multimodal distribution is observed. Generally, low polydispersity is preferred, especially for biomedical applications. The authors should explain in the text more convincingly, how high polydispersity would not be a barrier for medical applications.

2.       In Fig. 2, the nanoparticle size at 37 oC is almost 5 times larger than at room temperature for the HPC formulation. Since the body temperature is also 37 oC, the question of how the nanoparticles will be stored becomes important.

3.       The peptide design should rely on carefully chosen peptide sequence that can self-assemble into nanoparticles. This typically involves incorporating hydrophobic and hydrophilic regions within the peptide sequence to promote self-assembly. The peptide should also have sufficient solubility in the desired solvent or buffer. The authors may find a recent paper covering such aspects helpful: Prediction of Amphiphilic Cell-Penetrating Peptide Building Blocks from Protein-Derived Amino Acid Sequences for Engineering of Drug Delivery Nanoassemblies, J. Phys. Chem. B 2020, 124, 20, 4069–4078.

Author Response

The authors investigate the preparation of stable high concentration peptide nanoparticles. The subject involves several considerations to ensure their stability and prevent aggregation. Key factors that are considered are the peptide design, selection of the solvent, pH optimization, concentration optimization, temperature dependence, mixing techniques, prevention of contamination, stabilizing agents, and characterization by physicochemical methods. The particular novelty of the paper is the use of low shear resonant acoustic mixing for peptide nanoparticle preparation.

We thank the reviewer for their thoughtful comments and suggestions.  Please see our responses and revisions regarding each point below:

  1. The polydispersity data reported in Tables 1 and 2 appear to be very high. Even in some cases, a multimodal distribution is observed. Generally, low polydispersity is preferred, especially for biomedical applications. The authors should explain in the text more convincingly, how high polydispersity would not be a barrier for medical applications.

The data reported in Table 1 are the results from an initial screen to identify the optimal formulation stabilizer composition to generate stable nanoparticles.  As a result, some of the formulation compositions may result in larger particle sizes with wider polydispersities that are not as stable.  In this case, we selected the two samples for scale-up that gave the smallest average particle size (measured as an average of 10 different acquisitions).  The polydispersities for these samples reported in Table 2 indicate that the particle size populations are monomodal, with %Pd of 33.6 and 19.9 for the HPC-SL/SDS and PVP K29-32/SDS compositions respectively.  Samples with polydispersity <25% are considered monomodal with a fairly narrow particle size distribution, demonstrating that the PVP K29-32/SDS compositions exhibited more efficient and stable nanomilling.  However, as the reviewer noted, for further drug product optimization for biomedical applications, further nanomilling work would be conducted to improve this profile.

  1. In Fig. 2, the nanoparticle size at 37 oC is almost 5 times larger than at room temperature for the HPC formulation. Since the body temperature is also 37 oC, the question of how the nanoparticles will be stored becomes important.

The stability results reported in Figure 2 represent the stability of the nanosuspension formulations under different temperature conditions, including cold storage at 4 °C and accelerated conditions at higher temperature of 37 °C.  This was not intended to represent biorelevant or physiological stability but rather to understand the tendency of these formulations to aggregate over time.  Nevertheless, for the HPC formulation, the nanoparticles exhibit aggregation after 14 days, much longer than an injection would remain in the body, where the nanoparticles would be rapidly absorbed.

  1. The peptide design should rely on carefully chosen peptide sequence that can self-assemble into nanoparticles. This typically involves incorporating hydrophobic and hydrophilic regions within the peptide sequence to promote self-assembly. The peptide should also have sufficient solubility in the desired solvent or buffer. The authors may find a recent paper covering such aspects helpful: Prediction of Amphiphilic Cell-Penetrating Peptide Building Blocks from Protein-Derived Amino Acid Sequences for Engineering of Drug Delivery Nanoassemblies, J. Phys. Chem. B 2020, 124, 20, 4069–4078.

We thank the reviewer for their suggestion and have found their reference helpful.  In this paper, we have purposefully selected three peptides of diverse structural and molecular space in order to demonstrate the generality of the acoustic nanomilling method.  These considerations will certainly be important when optimizing this approach to new peptides.

Round 2

Reviewer 1 Report

Comments and Suggestions for Authors

The authors of the work have responded to all comments posted. I have two more comments. The error marked on the bar graphs either has a graphical problem, the error is larger than the bar shown there (Fig. 5), or it is not shown at all (Fig. 10.) Please check them.

I do not have access to SI to see the tables the authors mention in their responses to the comments.

Author Response

We thank the reviewer for their careful review of the updated manuscript.  Based on their comments, we have corrected figures 5 and 10---we have adjusted the vertical axis to ensure that the entirety of the data bar is shown, including the error bars.  We have also gone back to look at the raw DLS data and have excluded data that was not of sufficient quality (i.e., the signal to noise was too low, resulting in an autocorrelation curve that could not be fit accurately by either cumulants or regularization fit).  The updated data has been included in the above figures.  

Based on the reviewer's initial comments from their first review, the standard deviation for particle size have been included in Tables 1, 2, 3, 4, 5, and 6 with the particle size data methodology described in the Materials and Methods section 3.4.1.